# Potential Roles of microRNAs for Assessing Cardiovascular Risk in Pre-Eclampsia-Exposed Postpartum Women and Offspring

**DOI:** 10.3390/ijms242316842

**Published:** 2023-11-28

**Authors:** Nurul Iffah Mohd Isa, Saiful Effendi Syafruddin, Mohd Helmy Mokhtar, Shahidee Zainal Abidin, Farah Hanan Fathihah Jaffar, Azizah Ugusman, Adila A. Hamid

**Affiliations:** 1Department of Physiology, Faculty of Medicine, Universiti Kebangsaan Malaysia, Jalan Yaacob Latif, Bandar Tun Razak, Kuala Lumpur 56000, Malaysia; p130569@siswa.ukm.edu.my (N.I.M.I.); helmy@ukm.edu.my (M.H.M.); farahhanan@ukm.edu.my (F.H.F.J.); dr.azizah@ppukm.ukm.edu.my (A.U.); 2UKM Medical Molecular Biology Institute, Universiti Kebangsaan Malaysia, Jalan Yaacob Latif, Bandar Tun Razak, Kuala Lumpur 56000, Malaysia; effendisy@ppukm.ukm.edu.my; 3Faculty of Science and Marine Environment, University Malaysia Terengganu, Kuala Nerus 21030, Malaysia; shahidee.zainal@umt.edu.my

**Keywords:** pre-eclampsia, microRNA, postpartum, offspring, cardiovascular disease

## Abstract

Pre-eclampsia, which is part of the spectrum of hypertensive pregnancy disorders, poses a significant health burden, contributing to maternal and infant morbidity and mortality. Pre-eclampsia is widely associated with persistent adverse effects on the cardiovascular health of women with a history of pre-eclampsia. Additionally, there is increasing evidence demonstrating that offspring of pre-eclamptic pregnancies have altered cardiac structure and function, as well as different vascular physiology due to the decrease in endothelial function. Therefore, early detection of the likelihood of developing pre-eclampsia-associated cardiovascular diseases is vital, as this could facilitate the undertaking of the necessary clinical measures to avoid disease progression. The utilisation of microRNAs as biomarkers is currently on the rise as microRNAs have been found to play important roles in regulating various physiological and pathophysiological processes. In regard to pre-eclampsia, recent studies have shown that the expression of microRNAs is altered in postpartum women and their offspring who have been exposed to pre-eclampsia, and that these alterations may persist for several years. This review, therefore, addresses changes in microRNA expression found in postpartum women and offspring exposed to pre-eclampsia, their involvement in cardiovascular disease, and the potential role of microRNAs to be used as predictive tools and therapeutic targets in future cardiovascular disease research.

## 1. Introduction

Hypertensive disorders of pregnancy (HDPs) represent a significant health burden to both mother and child, leading to complications of approximately 10% of pregnancies worldwide [1]. According to the International Society for the Study of Hypertension in Pregnancy (ISSHP), the classification of HDPs includes pre-eclampsia, pre-eclampsia superimposed on chronic hypertension, gestational hypertension, chronic hypertension, and isolated office hypertension [2]. Among these types, pre-eclampsia is one of the best-known antenatal problems, affecting between almost 3% and 5% of all pregnancies, and is responsible for more than 500,000 and 70,000 foetal/neonatal and maternal deaths, respectively, annually [2,3,4]. Pre-eclampsia is clinically diagnosed when de novo hypertension develops after the 20th week of pregnancy, followed by significant proteinuria, dysfunction of organs such as the kidney and liver, haematological and neurological problems, and foetal growth restriction [2]. Women and infants exposed to pre-eclampsia are associated with several adverse outcomes, particularly increased susceptibility to developing cardiovascular disease (CVD) in the future [5,6,7]. Several cohort studies have reported positive correlations between women with a history of pre-eclampsia and mortality from cardiovascular disease and hypertension in the offspring of women with pre-eclampsia [8,9,10,11]. Therefore, it is crucial to determine the risk of postpartum women and their offspring exposed to pre-eclampsia for CVD as early as possible.

microRNAs (miRNAs) have been widely implicated in the pathogenesis of various diseases. Due to their abundance/presence and stability in biological fluids, substantial research efforts have been made to develop miRNAs as potential non-invasive biomarkers for disease diagnosis and prognosis [12]. miRNAs are short, non-coding RNAs that regulate gene expression post-transcription process. They are potent gene expression regulators that are involved in various biological functions such as cell proliferation, differentiation, and death [13]. Extensive research has shown that miRNA expression is also altered in CVD [14,15,16]. Several studies have shown that aberrant cardiac gene expression leads to morphological and functional changes in the heart such as hypertrophy, irregular conduction, decreased cardiomyocyte contractility and survival, and disruption of vascular homeostasis [17]. This has led to recent research focusing on miRNAs as biomarkers for diseases such as cardiovascular disease in people at risk, including postpartum women exposed to pre-eclampsia and their offspring. This research interest is increasing, as the role of miRNAs in postpartum women and their offspring exposed to pre-eclampsia is not well understood. Therefore, this review provides insight into various findings of miRNAs expressed in postpartum women with a history of pre-eclampsia and their offspring, as well as the potential role of miRNAs in the development of CVD.

## 2. Pathogenesis of Pre-Eclampsia

The initial stage of pre-eclampsia begins with abnormal development of the blood vessels in the spiral arteries of the mother’s uterus [18]. In a normal pregnancy, the cells that form the outer layer of the blastocyst, called trophoblasts, invade the myometrium of the uterus and destroy the smooth muscle of the spiral arteries. This allows the spiral arteries to dilate and provides smooth flow of blood with little resistance to the placenta [19]. However, in pre-eclampsia, the spiral arteries remain unchanged because the trophoblasts do not invade the smooth muscle of the spiral arteries, resulting in increased resistance to blood flow as the smooth muscle of the spiral arteries continues to constrict [20]. In a study by Lyall et al., a reduction in muscle disruption in the spiral arteries and fewer intramural extravillous cytotrophoblasts on the myometrial vessels were found in pre-eclampsia patients compared to the control group [21]. A study by Zhou et al. reported that limited invasion of cytotrophoblasts happens due to the aberrant cytotrophoblast gene expression known as *SEMA3B*, as this gene is upregulated in patients with pre-eclampsia [22]. This gene encodes for the protein semaphorin 3B, which is able to cause the disruption of angiogenesis through binding with vascular endothelial growth factor (VEGF) family members and promotes the apoptosis of cells [23]. Invasion of cytotrophoblasts was reduced by 60% as the rate of cytotrophoblast apoptosis increased [22]. Incomplete conversion of the spiral arteries leads to a reduction in uterine blood flow and foetal blood supply and has been associated with foetal growth restriction, placental abruption, and spontaneous premature rupture of the membranes, in addition to a number of other severe obstetric disorders [24,25]. Atherosis, sclerotic narrowing of arteries and arterioles, fibrin deposition, and infarcts are found to be common pathological features in pre-eclamptic placentas. These pathological abnormalities are all associated with placental hypoperfusion and ischaemia and appear to be correlated with the severity of pre-eclampsia [26].

Angiogenic factors such as the Fms-like tyrosine kinase receptor (Flt-1) or VEGF receptor 1 (VEGFR-1) and placental growth factor (PlGF) are thought to play an important role in regulating the vascular development of the placenta, and the imbalance of these angiogenic factors contributes to pre-eclampsia [27]. PlGF promotes angiogenesis by binding to the Flt-1 receptor, allowing VEGF to bind to VEGFR [28]. The overexpression of soluble Flt-1 (sFlt-1) is associated with a reduction in cytotrophoblast invasiveness and a reduction in PlGF and VEGF signalling, as these factors are all expressed by invasive cytotrophoblasts [29]. This is due to the fact that sFlt-1 acts antagonistically and prevents VEGF and PlGF from binding to Flt-1 by first binding these factors in the circulation [30]. Zhou et al. found that VEGF family members play an important role in regulating cytotrophoblast survival and that VEGF members are dysregulated in severe pre-eclampsia [29]. In addition, sFlt-1 is more highly expressed in the bloodstream, resulting in lower VEGF and PIGF levels in pre-eclampsia compared to normal pregnancy [31,32,33].

Syncytiotrophoblast-derived extracellular vesicles (SDEVs), derived from apoptotic or activated syncytiotrophoblast cells, are increased in expression in pre-eclampsia patients because these vesicles are responsible for the transport of sFlt-1 along with proinflammatory cytokines and coagulation factors, leading to worsening maternal response in terms of vascular function [34], inflammation [35], and coagulation [36], and transforming the physiological response of a healthy pregnancy into pre-eclampsia [37].

However, the pathogenesis described above can only be applied to a specific subtype of pre-eclampsia, namely the early-onset pre-eclampsia described by Huppertz (before the 34th week of pregnancy), which occurs in only 5 to 20% of all cases of pre-eclampsia [38]. Inadequate placentation and foetal growth restriction are associated with the early-onset form, while maternal factors are thought to be the cause of the late-onset form without having any effect on the placenta [39]. Huppertz added another important point in describing the development of pre-eclampsia, as it may be caused by the mother’s susceptibility to the condition. In addition to the condition of the placenta, maternal environmental factors such as genetic predisposition, family history, and cardiovascular health must also be considered [38]. Environmental changes can influence epigenetic changes, including DNA methylation, histone modification, and non-coding RNA, which may ultimately contribute to pre-eclampsia [40].

## 3. Risk of Cardiovascular Disease in Postpartum Women Exposed to Pre-Eclampsia

Pre-eclampsia has been found to be associated with a higher likelihood of future heart failure, ischaemic and hypertensive heart disease, and death related to CVD compared to normal pregnancy [41]. In the UK Biobank cohort study, women with HDPs were found to have a higher arterial stiffness index and a higher prevalence of chronic hypertension, associated with a higher incidence of many CVD-related conditions, including coronary artery disease, aortic stenosis, heart failure, and mitral regurgitation compared to women without HDPs [42]. In addition, women with preterm pre-eclampsia, and to a lesser extent term pre-eclampsia, have a higher risk of asymptomatic left ventricular dysfunction/hypertrophy and essential hypertension within 1 to 2 years postpartum [41]. It was found that 65% of cases of CVD occurred in women affected by pre-eclampsia, and within 1 year of the first index pregnancy, the incidence became marked [43]. In addition, women who have experienced pre-eclampsia have a nearly four-fold higher risk of hypertension and a nearly two-fold higher risk of fatal and non-fatal ischaemic heart disease, stroke, and venous thromboembolism later in life [44].

The mechanisms underlying the increased risk of cardiovascular disease in women with a history of pre-eclampsia are not yet fully elucidated, but endothelial dysfunction associated with atherosclerosis persists in women for many years after a pre-eclamptic pregnancy [45]. Significantly elevated levels of vascular cell adhesion molecule 1 (VCAM-1) and intercellular cell adhesion molecule 1 (ICAM-1) have been found in women with a history of pre-eclampsia [46]. In addition, groups of women who had pre-eclampsia were sensitive to angiotensin II and salt, which resulted in a significant increase in aldosterone, sFlt-1, and blood pressure after a low-salt diet was introduced [47]. Some studies have shown that the imbalance of the angiogenic factor sFlt-1 during pregnancy puts women with pre-eclampsia at increased risk of CVD after birth. The increase in sFlt-1 affects the release of endothelin-1, a potent vasoconstrictor, and also inhibits the phosphorylation of endothelial nitric oxide synthase (eNOS) in pre-eclamptic women, which may increase sensitivity to vasoconstriction and sensitivity to angiotensin II, both of which may eventually lead to hypertension [48,49]. A study by Garrido-Gimenez found a negative correlation between sFlt-1 and high-density lipoprotein, and sFlt-1 was also positively related to carotid intima–media and left ventricular posterior wall thickening 12 years after index pregnancy [50]. Therefore, sFlt-1 is often used as a biomarker to detect CVD [51,52,53].

As the relationship between pre-eclampsia and cardiovascular disease is complex, it is important to consider several factors that may contribute to this relationship. While a causal relationship between pre-eclampsia and cardiovascular disease risk has previously been emphasized, it is important to recognize alternative explanations, such as common underlying genetic abnormalities.

One plausible hypothesis is that both pre-eclampsia and the subsequent cardiovascular diseases are an expression of the same genetic predisposition or susceptibility. In this context, pregnancy could be seen as a “stress test” for the cardiovascular system, revealing vulnerabilities that may contribute to both conditions. Individuals who experience pre-eclampsia may have an increased predisposition to cardiovascular problems, which could explain the later increased risk [54].

## 4. Risk of Cardiovascular Disease in Offspring Exposed to Pre-Eclampsia

Over the years, extensive studies have been carried out to elucidate the consequences of pre-eclampsia on the later development of cardiovascular diseases in the exposed infants. Women with pre-eclampsia are more likely to deliver their babies prematurely than women who experienced a healthy pregnancy, as reported in a population-based control study conducted by Davies et al. [55]. Premature-born babies are at risk of having CVD compared to those born at term [56].

A prospective study (HUNT) conducted in Norway found that the offspring of mothers with hypertensive pregnancies, including pre-eclampsia, had higher systolic and diastolic blood pressure and 35% higher levels of c-reactive protein (CRP) than the children of women with normal pregnancies [57]. CRP, an inflammatory molecule, is often used as a marker for cardiovascular risk assessment, particularly for coronary atherosclerosis, as this molecule is produced in the liver following the release of proinflammatory cytokines such as interleukin (IL)-1β, IL-6 and tumour necrosis factor α (TNF-α) [58]. Arroyo-Espliguero et al. also mentioned that inflammation plays a crucial role in atherogenesis, the process of plaque formation in the arterial wall, so using inflammatory markers to predict cardiovascular disease is useful, especially in people who are at risk for this disease [58]. This has been demonstrated by the high level of CRP in patients with coronary artery disease, and the level of this protein increases with the severity of the disease [59]. Angiogenic markers such as soluble endoglin (sENG) and sFlt-1 have also been used as indicators of CVD, as sENG and sFlt-1 were found to be positively correlated with the offspring of mothers with pregnancy-induced hypertension since these two markers were significantly elevated compared to controls [60].

Apart from inflammatory and angiogenesis markers, the lipid profile of infants born to mothers with pre-eclampsia differed from that of normal offspring, as it was observed that serum triglycerides and non-high-density lipoprotein (HDL) were significantly elevated, in addition to a significant decrease in serum HDL levels [57,61,62]. HDL is important for reverse cholesterol transport (RCT), which transports excess cholesterol from the peripheral arteries back to the liver where it is disposed of, and has beneficial properties for improving cardiovascular function [63].

In addition to differences in inflammatory and angiogenesis markers and lipid metabolism in the blood, several studies have reported changes in cardiac structure in the children of women with pre-eclampsia. It was found that the carotid intima–media of the offspring of hypertensive pregnant women were thicker, the arteries were stiffer, and the dilation of the arteries for blood flow decreased [64]. In addition, the left ventricular mass (LV) of the offspring with pre-eclampsia was higher compared to normal, as measurements of relative wall thickness (RWT) and left ventricular mass index (LVMI) were greater and left diastolic end volume (LDEV) decreased [65]. LVMI is often used to assess LV hypertrophy caused by arterial hypertension, and abnormal RWT findings indicate LV remodelling, which is an early sign of hypertensive heart disease [66]. The increase in blood pressure contributes to the increase in LV and causes LVH and changes in LV geometry due to chronic haemodynamic stress and central pressure overload [67].

Although maternal factors, including pre-eclampsia, may play a role, they are not the only determinants of cardiovascular risk in offspring. Common genetic abnormalities can contribute significantly to cardiovascular disease risk in offspring. To various extents, coronary artery disease and other regular, complex cardiovascular disorders are heritable; even the environment and lifestyle are able to influence the development of cardiovascular disease [68,69,70].

Furthermore, foetal growth restriction (FGR), which often occurs in early PE, is an additional risk factor. FGR can have its own effects on the child’s health, including an increased risk of cardiovascular problems later in life. Cardiovascular remodelling in small-for-gestational-age (SGA) infants persisted after 6 months of delivery, where blood pressure in this group of infants increased with the thickening of the aortic intima–media [71]. In addition, the cardiovascular morbidity of dichorionic–diamniotic twins with FGR was found to be higher than non-FGR twins in a population-based retrospective cohort study [72].

## 5. microRNA Biogenesis

miRNAs are small non-coding RNAs with the length of 20–22 nucleotides that repress their respective mRNA targets either by inducing mRNA degradation or blocking the translation process [73]. miRNAs are crucial for human development as they are involved in various biological processes, such as cell proliferation, differentiation, and death, as well as modulating the immune responses [73]. Most miRNAs are produced by transcribing DNA sequences into primary miRNAs, which are processed into precursor miRNAs and eventually mature miRNAs [74].

miRNA biosynthesis can occur either via canonical or non-canonical pathways. The canonical pathway, the most common pathway for miRNA synthesis, begins in the nucleus when RNA polymerase II starts transcribing the miRNA genes that lead to the synthesis of primary miRNA (pri-miRNA) [75,76] (Figure 1). Most miRNA genes are located in the intragenic region of coding/non-coding genes, in repeated regions in genomes, or in untranslated regions (UTRs) [77]. A microprocessor complex then processes the pri-miRNAs produced to form precursor miRNAs(pre-miRNAs) [78]. The microprocessor complex consists of DROSHA, which belongs to the RNase type endonucleases III, and DiGeorge syndrome critical region 8 (DGCR8). DGCR8 structure can recognise the hairpin structure of pri-miRNAs, leading to the cleavage of the hairpin structure by DROSHA, which is called pre-miRNA [79]. Subsequently, the pre-miRNAs are transported to the cytoplasm by a complex of exportin-5 (XPO-5) and RAS-related nuclear protein-guanosine-5′-triphosphate-ase (Ran-GTPase) [80]. XPO-5 has the ability to protect nuclear pre-miRNAs from degradation [81].

In the cytoplasm, the terminal loop structure of the pre-miRNA is digested by another RNase III, known as Dicer, converting these pre-miRNAs into mature duplex miRNAs [82,83]. The process of cleavage of the terminal loop of pre-miRNA is assisted by another protein called TAR RNA binding protein (TRBP) and kinase-R activating protein (PACT), as these proteins enhance the stability and processing activity of Dicer, with TRBP increasing the binding affinity and cleavage precision of Dicer and PACT acting as a cofactor to facilitate processing by Dicer [84,85]. Once the miRNA duplex is formed, the duplex is loaded and unwound into the Argonaute (Ago) proteins, and only one strand, the guide strand, is selectively anchored to AGO to become the miRNA-induced silencing complex (miRISC), while the other strand, known as the passenger strand, is ejected and degraded by AGO [86]. The miRISC binds to reverse complementary sequences between the 3′-UTR of target mRNAs [87]. The binding helps destabilise and repress the expression of the target mRNA as AGO recruits the TNRC (trinucleotide repeat containing 6) protein by inducing its decapping and deadenylation [88]. Ago2 is the only Ago subgroup that has the ability to cut the target mRNA [89].

Recently, several miRNAs that are produced contrary to the canonical biogenesis pathway have been discovered because they bypass one or more steps in this pathway and originate from multiple sources. These miRNAs are referred to as non-canonical miRNAs [90]. Some non-canonical pri-miRNAs can be encoded in the introns of the coding genes known as mirtrons [91]. Mirtrons are first processed by the nuclear splicing machinery and form stable hairpins with a short stem [92]. DROSHA/DGCR8 is unable to process these shorter hairpin structures; instead, they are debranched by debranching enzyme 1 (DBR1) [93]. In addition, small nucleolar RNAs (snoRNAs) serve as a source of pre-miRNAs that are not processed by the microprocessor [91]. For example, it was discovered that ACA45, a human snoRNA, is independent of DROSHA/DGCR8 processing and requires only one DICER. Its precursor structure resembles two pre-miRNA-like hairpins connected by a hinge [94]. MiRNAs generated from short hairpin RNA (shRNA) lack the microprocessor binding sequence and are indep endent of DGCR8 but dependent on DICER. For example, the mir-320 and mir-484 transcripts are capable of forming hairpin structures and are, therefore, referred to as endogenous shRNAs [95,96]. Non-canonical miRNAs can also be obtained from the stem of transfer RNAs (tRNAs) using DICER or angiogenin (ANG), as it cleaves the stem into tRNA-derived RNA (tDR) fragments [97]. It has been discovered that non-canonical miRNAs are involved in a variety of biological functions, including the immune response, and also play an important role in disease development [90,98].

## 6. microRNA Dysregulation in Postpartum Women Exposed to Pre-Eclampsia

A method for determining CVD risk in women exposed to pre-eclampsia has been introduced, incorporating a machine learning tool for prediction [99]. However, the process of implementing machine learning tools is complicated and requires a complete data set and a long follow-up period for patients [99]. Therefore, the use of miRNA as a predictive tool was introduced as miRNA met most of the requirements to be an excellent biomarker, including accessibility, high sensitivity, and specificity [100]. In addition, the application of miRNAs as a predictive tool is supported through the integration of computational analysis in order to improve the accuracy of the prediction [101]. miRNAs have been used widely as a diagnostic tool of various diseases such as cancer [102,103], Alzheimer’s disease [104], spinal cord injury [105], and chronic obstructive pulmonary disease [106]. It has been demonstrated that changes in the expression of miRNAs in women who experienced a pre-eclamptic pregnancy persist over the years following delivery (Table 1). In a study by Murphy et al., hsa-miR-221-3p was the only miRNA that was significantly altered in pre-eclampsia, as the expression of this miRNA is reduced in postpartum pre-eclampsia as compared to woman with normal pregnancy after one year [107]. A significant reduction in the miRNA-221/222 family, which consists of miR-221-3p and miR-222-3p, consisting of miR-221-3p and miR-222-3p, is associated with patients with dilated cardiomyopathy and patients with aortic stenosis with severe fibrosis [108]. Furthermore, the inhibition of miR-221/222 exacerbates angiotensin-induced left ventricular dysfunction and dilatation II, suggesting greater susceptibility to pressure-overload-induced heart failure in an in vivo study [108]. In addition, miR-221/222 has been found to be downregulated in endothelial cells exposed to inflammatory stimuli such as IL-3 and basic fibroblast growth factor (bFGF), and overexpression of miR-222 can challenge vascular remodelling modulated by STAT5A [109].

A significant reduction in three circulating miRNAs, miR-126-3p, miR-146a-5p, and miR-122-5p, was found in women with a history of premature acute coronary syndrome (ACS) and pre-eclampsia [110]. miR-126 has the ability to enhance the action of angiogenic factors, including VEGF and fibroblast growth factor (FGF), and stimulate blood vessel formation by inhibiting the angiogenic signalling inhibitor Spred-1 [111]. Expression of miR-126 was found to be suppressed in patients with cardiovascular disease, with this miRNA being downregulated in patients with coronary artery disease [16]. Furthermore, reduced expression of miR-126 in patients with pulmonary arterial hypertension contributed to right ventricular failure due to low angiogenesis activity [112] and led to the development of diabetic cardiac microangiopathy [113]. In vivo studies found that the downregulation of miR-146a caused cardiac contraction dysfunction through tyrosine kinase inhibition [114]. A significant reduction in miR-122 expression has been found in patients with coronary artery disease [115], and miR-122 has been linked to the regulation of lipid and cholesterol metabolism [116].

A study by Abad found that the expression of miR-103a-3p was downregulated in women with a history of pre-eclampsia compared to women with a healthy pregnancy [117]. The increased expression of miR-103-3a is important as it is able to protect cardiomyocytes from apoptosis and autophagy under hypoxia/reoxygenation by suppressing *Atg5* expression and disrupting the Bcl-2-Beclin-1 interaction, which is involved in the regulation of autophagy [118]. RUNX2, a transcription factor for osteogenesis, increases its expression in calcified atherosclerotic plaques and plays an important role in vascular calcification [119]. However, He et al. found that upregulation of miR-103a is able to reduce the expression of RUNX2, indicating the important function of this miRNA in preventing calcification in blood vessels [119].

Upregulation of three miRNAs, namely miR-1-3p, miR-23a-3p, and miR-499a-5p, was observed in women with a history of complicated pregnancies who have vascular endothelial dysfunctions, including pre-eclampsia [120]. The levels of miR-1 and miR-499 were expressed highly in patients with acute myocardial infarction [121]. miR-1 has been shown to be involved in cardiovascular disease. Terentyev et al. found that increased expression of miR-1 enhances the excitability of cardiac contraction by increasing the release of calcium ions (Ca^2+^) in cardiac myocytes, and this situation is often found in cardiac arrhythmias [122]. High levels of miR-1 have been associated with a reduction in the number of Kir2.1, a channel that controls the movement of potassium ions (K^+^), as this miRNA is involved in the downregulation of the *KCNJ2* gene, which encodes Kir2.1, and this leads to an imbalance between Ca^2+^ and K^+^ levels, causing cardiac rhythm disturbances [123]. In patients with coronary artery disease, miR-23a has been found to be upregulated, and the increase in miR-23a levels causes the number of endothelial progenitor cells (EPCs) to be suppressed by miR-23a due to epidermal growth factor receptor (EFGR) activity [124]. Circulating miR-499a-5p has been linked to hypertrophic cardiomyopathy, i.e., hypertrophy of the left ventricle, as increased expression of miR-499a-5p is associated with high expression of the mutant *MYH7* gene [15].

Another study by Hromadnikova et al. to profile postpartum women with pre-eclampsia discovered that six miRNAs, namely miR-17-5p, miR-20b-5p, miR-29a-3p, miR126-3p, miR-133a-3p, and miR-1-3p, were upregulated and miR-130b-3p was downregulated in peripheral blood samples from pre-eclampsia [125]. miR-17-5p has been found to be involved in apoptosis in myocardial infarction, as downregulation of miR-17-5p leads to an increase in the anti-apoptosis protein Bcl-2 and decreases the expression of the apoptotic proteins Bax, caspase-3, and caspase-9 [126,127]. Induction of cardiac hypertrophy was found after miR-20b-5p expression was increased by cytosolic Ca^2+^ overload [128]. Increased expression of miR-126-3p was found in hypertensive patients with albuminuria [129]. miR-130b-3p was observed to promote angiogenesis in human umbilical vein endothelial cells (HUVEC) and negatively regulate *PTEN* expression [130]. *PTEN* downregulates phosphatidylinositol-3 kinase (PI3K) system, a key regulator for cell growth and survival in cardiac and endothelial cells, and PTEN loss of function improves the function of the heart by promoting cardiomyocyte proliferation, decreasing cardiac fibrosis, and also preventing apoptosis due to ischaemic stress [131,132].

**Table 1 ijms-24-16842-t001:** Summary of miRNA expression in postpartum women with history of pre-eclampsia.

miRNA	Regulation	Role in CVD	Sample	Reference
miR-221-3p	Downregulated	Cell proliferation	Plasma	[107]
miR-126-3p	Downregulated	Angiogenesis		[110]
miR-146a-5p	Downregulated	Angiogenesis		
miR-122-5p	Downregulated	Fat metabolism		
miR-103a-3p	Downregulated	Anti-apoptosis and autophagy		[117]
miR-1-3p	Upregulated	Action potential regulation	Whole Peripheral Blood	[120,125]
miR-23a-3p	Upregulated	Apoptosis, cell regeneration		[120]
miR-499a-5p	Upregulated	Cell proliferation		
miR-17-5p	Upregulated	Apoptosis		[125]
miR-20b-5p	Upregulated	Cell proliferation, migration		
miR-29a-3p	Upregulated	Cell migration		
miR-126-3p	Upregulated	Angiogenesis		
miR-133a-3p	Upregulated	Anti-angiogenesis		
miR-130b-3p	Downregulated	Angiogenesis		

## 7. microRNA Dysregulation in Offspring Exposed to Pre-Eclampsia

The use of miRNA to predict CVD risk is not limited to mothers, but can also predict CVD risk in offspring exposed to pre-eclampsia, as discovered in several studies (Table 2). By assessing miRNA expression using cord blood, miR-26a-5p, miR-145-5p, miR-574-3p, miR-195-5p, miR-199a-5p, and miR-221-3p were found to be downregulated in severe pre-eclampsia and miR-92a-3p expression was increased in mild pre-eclampsia [133]. Hromadnikova et al. also found that the expression of miR-499a-5p was increased in placental tissue exposed to pre-eclampsia [134]. miR-26a-5p was found to be able to protect the heart from myocardial ischaemia/reperfusion injury. Features of myocardial ischaemia/reperfusion injury include cardiomyocyte loss, microvascular damage and inflammation by suppressing *PTEN* expression, and activating the PI3K/AKT pathway to increase viability and prevent cardiomyocyte apoptosis [135]. The excessive accumulation of extracellular matrix and proliferation of cardiac fibroblasts, which can lead to heart failure, malignant arrhythmias, and sudden death in hypertensive cardiac fibrosis, was inhibited by the increased expression of miR-26a through inhibition of cardiac fibroblast proliferation via the 2/p21 pathway [136]. Cardiomyocyte apoptosis was prevented by upregulating miR-145-5p and miR-195-5p by targeting ROCK1 and inhibiting TGF-β1/Smad3 pathway activation, respectively [137,138]. miR-574, which comprises miR-574-3p and miR-574-5p, is actively involved in the maintenance of mitochondrial function and protects the heart from pathological remodelling caused by cardiac stress by regulating *FAM210A* expression [139]. miR-199a is regularly implicated in STAT3 signalling, and miR-199a-5p has been found to be downregulated during chronic hypoxia when STAT3 signalling is activated [140]. miR-221-3p can inhibit angiogenesis, as overexpression of miR-221-3p can inhibit proliferation, migration, and strand formation of endothelial cells in vitro by targeting HIF-1α [141]. miR-92a-3p was found to be associated with the activation of NF-ĸB, as the expression of this miRNA positively correlated with the expression of NF-ĸB along with other inflammatory markers such as monocytochemotactic protein-1 (MCP-1), endothelin-1 (ET-1), and intercellular adhesion molecule-1 (ICAM-1). This proves that miR-92a may play an essential role in the development of CVD through the activation of NF-ĸB and downstream inflammatory pathways [14].

Changes in the expression of miRNA in offspring exposed to PE can also be studied using umbilical cord-derived endothelial cells. Zhou et al. discovered dysregulation of miR-29a/c-3p in human umbilical vein endothelial cells (HUVECs) in pre-eclampsia, where downregulation of miR-29a/c-3p may impede foetal endothelial cell migration by disrupting FGF2-stimulated PI3K-AKT1 signalling [142]. In addition to miR-29a/c-3p, miR-146a was also significantly expressed in hypertensive HUVECs [143]. MiR-146a has been found to play a role in inflammation [144]. In an in vivo study, treatment of adult mouse cardiomyocytes with miR-146a-5p packaged in extracellular vesicles was found to activate the expression of cytokines such as CXCL2, IL-6, and TNF-α, as well as innate immune cells, namely CD45+ leukocytes, Ly6Cmid+ monocytes, and Ly6G+ neutrophils [145]. In contrast to HUVEC, RNA sequencing of endothelial progenitor cells (EPCs) showed that hsa-miR-1270 expression decreases in pre-eclampsia [146]. Furthermore, Brodowski et al. discovered that inhibition of hsa-miR-1270 knockdown in EPCs impaired the tube-forming ability and chemotactic motility of the cells [146].

The changes in miRNA expression cannot only be detected directly after birth using samples from the placenta. It has even been found that the changes in miRNA expression persist for years in the offspring of mothers with pre-eclampsia. This was demonstrated in a study by Hromadnikova et al., who found that the expression of miR-133a-3p, miR-1-3p, miR-103a-3p, and miR-20a-5p was increased and the expression of miR-342-3p was significantly decreased in the offspring of pregnancies with pre-eclampsia [147]. Overexpression of miR-103a-3p was found in patients with hypertensive nephropathy (HN), and mice infused with angiotensin II had higher levels of circulating miR-103a-3p; this miRNA has the ability to suppress anti-inflammatory SNRK expression in glomerular endothelial cells. The reduction of SNRK levels in glomeruli of HN patients and mice infused with angiotensin II occurred through the activation of NF-κB/p65 pathway, which eventually led to renal inflammation and fibrosis [148]. miR-20a-5p is another miRNA that regulates the inflammatory process, as it was discovered that TGF-β signalling plays a crucial role in suppressing inflammation and downregulation of miR-20a-5p can alter inflammation and fibrosis in the liver, as this miRNA is an important regulator of inflammation-induced liver fibrosis [149]. In CVD, miR-342-3p exerted an anti-inflammatory effect, as the expression of miR-342-3p was reduced in type 1 diabetes mellitus, while the expression of inflammatory markers IL-7, IL-8, and TNF-α was increased [150].

**Table 2 ijms-24-16842-t002:** Summary of miRNA expression in offspring exposed to pre-eclampsia.

miRNA	Regulation	Role in CVD	Sample	Reference
miR-26a-5p	Downregulated	Apoptosis inhibitor, autophagy regulation	Cord Blood	[133]
miR-145-5p	Downregulated	Apoptosis inhibitor		
miR-574-3p	Downregulated	Regulation of mitochondrial function		
miR-195-5p	Downregulated	Apoptosis inhibitor		
miR-199a-5p	Downregulated	Induced cell proliferation, anti-inflammation		
miR-221-3p	Downregulated	Anti-angiogenesis		
miR-92a-3p	Upregulated	Inflammation		
miR-499a-5p	Upregulated	Cell proliferation	Placenta	[138]
miR-29a/c-3p	Downregulated	Cell migration	HUVEC	[142]
miR-146a	Upregulated	Inflammation		[143]
miR-1270	Downregulated	Cell migration	EPC	[146]
miR-133a-3p	Upregulated	Anti-angiogenesis	Whole Peripheral Blood	[147]
miR-1-3p	Upregulated	Action potential regulation		
miR-103a-3p	Upregulated	Apoptosis, autophagy		
miR-20a-5p	Upregulated	Inflammation		
miR-342-3p	Downregulated	Anti-inflammation		

## 8. Limitations and Future Directions

Several miRNAs with the potential to be involved in CVD were found to be upregulated or downregulated in postpartum women exposed to pre-eclampsia and their offspring. The advantage of using miRNA as a predictive tool for CVD detection is that it is sufficiently stable when the sample is properly handled, and detection of miRNA is rapid and non-invasive [151]. Although the expression of miRNAs in women and offspring exposed to pre-eclampsia differs from that of women and children from normotensive pregnancies, the involvement of several miRNAs in cardiovascular disease is still unclear, and surprisingly, some of these miRNAs have even been shown to be beneficial in combating cardiovascular disease, depending on their expression in pre-eclampsia. For example, overexpression of miR-133a significantly reduced cardiac fibrosis and disrupted the phosphorylation of ERK1/2 and SMAD-2 [152]. miR-133a also has an anti-angiogenic effect, as expression of this miRNA reduced VEGF-induced proliferation of endothelial cells in HUVEC by also targeting ERK1/2 phosphorylation [153]. In addition, miR-103-3p was found to play an important role in preventing apoptosis and autophagy of cardiomyocytes under hypoxia/reoxygenation, as inhibition of miR-103a-3p leads to *Atg5* being able to promote apoptosis and autophagy in cells [118]. Therefore, further studies on the contribution of miRNAs involved in CVD need to be conducted, as this may help to improve our understanding of the role of miRNAs in the detecting CVD, especially in people at risk for this disease. For example, bioinformatics tools and databases could be used to analyse and predict the relationship between the expressed miRNA and its target gene. However, it cannot be denied that miRNA expression expressed in postpartum women with pre-eclampsia and their offspring will serve as a benchmark for them to maintain their health and for doctors to predict disease prognosis.

In addition to alterations in microRNA expression, epigenetic changes such as DNA methylation, histone modifications, and non-coding RNA expression that occur during the development of pre-eclampsia may also be considered in influencing the development of cardiovascular disease, as it has recently been discovered that epigenetic changes are involved in the pathogenesis of pre-eclampsia, as numerous genes are dysregulated in pre-eclampsia [154]. Even significant DNA alterations have been found mainly in genes related to inflammation and lipid metabolism, suggesting that epigenetic abnormalities can be detected early in pre-eclamptic infants [155].

Thus, in summary, miRNA expression can be used to determine CVD risk in women and their offspring who have been exposed to pre-eclampsia, but a deeper understanding of the role of the miRNAs involved and a consideration of the epigenetic influence in the development of the disease are required.

## Figures and Tables

**Figure 1 ijms-24-16842-f001:**
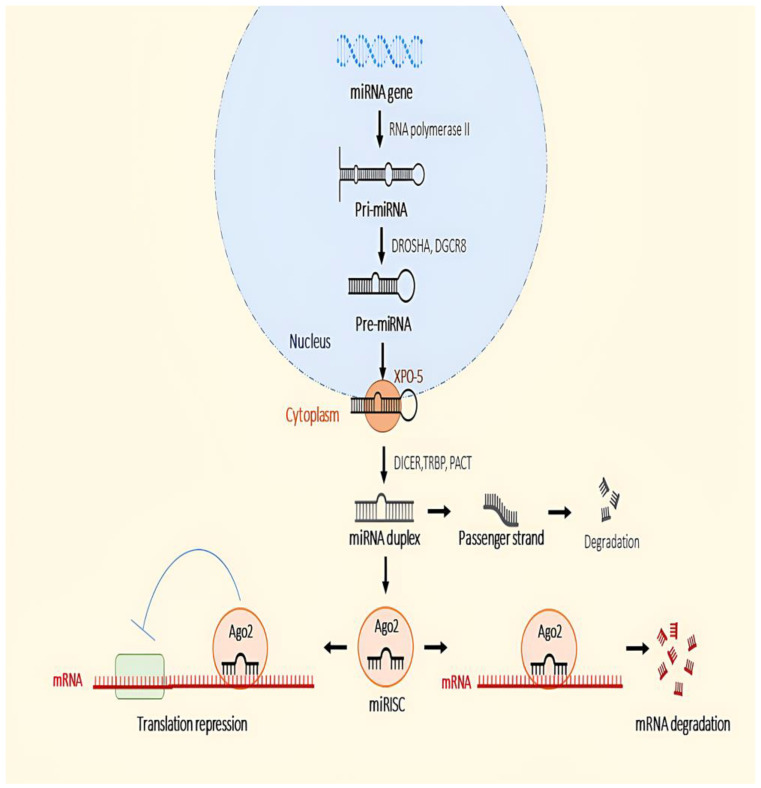
Canonical pathway in the biogenesis of miRNA.

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
