# Peer review of "Potential Roles of microRNAs for Assessing Cardiovascular Risk in Pre-Eclampsia-Exposed Postpartum Women and Offspring"

_ijms, 2023, doi:10.3390/ijms242316842_

Round 1
Reviewer 1 Report
Comments and Suggestions for Authors
I find this paper of great value, with information concerning the basal knowledge of preeclampsia and its consequences for the mother and her baby later in life.
The authors describe the pathophysiology of preeclampsia from the very beginning, the trophoblast invasion into the spiral arteries, through all cytokines that are involved in this process. The microRNA are the novel particles that are taken into account in the role of preeclampsia development. In my opinion, it is a valuable part of this study.
I have no other comments on this study
Author Response
Thank you for your comments. We appreciate you taking the time to read and review this paper.Reviewer 2 Report
Comments and Suggestions for Authors
This is an interesting review article on molecular insights of pre-eclampsia. Pre-eclampsia is associated with persistent adverse effects on maternal and neonatal cardiovascular health. Altered cardiac structure and function, as well as different vascular physiology due to the decrease in endothelial function are the main patways involved. Utilization of microRNAs as biomarkers is currently on the rise as microRNAs have been found to play important roles in regulating various physiological and pathophysiological processes and concerning pre-eclampsia, recent studies have shown that the expression of microRNAs is altered. This review addresses changes in microRNA expression found in postpartum women and offsprings of pre-eclamptic women, their involvement in cardiovascular disease, and potential role of microRNAs to be used as predictive tools and therapeutic targets in the future. Review of literature is adequate, methodology and analysis are valid, results are well-presented and conclusions are supported by the results. I think this paper adds to the existing literature and follows the new trend of advanced genetic analysis in fetal medicine.
Author Response
Thank you for your comments and the time you have taken to review this paper.Reviewer 3 Report
Comments and Suggestions for Authors
GENERAL REMARKS
This is a well written review of the role of micro RNA in PE and CV disease.
The authors seem to have the conception that PE causes an increase in CV disease risk and alterations in micro RNA, while it is more likely that the same underlying (genetic) abnormalities both cause PE and later CV disease risk, although epigenetic changes cannot be excluded. This should be made clear.
ABSTRACT
Line 16: Here you state that risk of CV disease is a result of PE, while it is more likely that both are the result of the same underlying genetic abnormalities. If you prefer not to change this sentence you should provide proof for your opinion.
INTRODUCTION
Line 40: The incidence of PE is appr 3-5% and not 15%. Ref 3 is a review and cites other papers, Ref 4 does not specify an incidence. You should get a better reference for your statement.
Line 45: It is more appropriate to cite here the ISSHP definition and not a local definition.
PATHOGENESIS
You describe the common conception of abnormal placentation leading to PE. However, this only applies to early PE with FGR. Furthermore, the association of PE with biomarkers or genetic upregulation might be sequential and not causal.
CARDIOVASCULAR DISEASE
Here, as in your abstract, you state that risk of CV disease is a result of PE, while it is more likely that both are the result of the same underlying genetic abnormalities. Some conceive pregnancy as a “stress test” for the cardiovascular system. Those that “fail this test” and develop PE have also an increased risk for later CV disease. You should at least describe this as a possible explanation for the association of PE and later CV disease.
RISK OFCV IN OFFSPRING
Here (again) your text seems to imply that PE I the mother is the cause of CV disease risk in the offspring, while it is more likely that common genetic abnormalities are the underlying cause. Additionally FGR occurs often in early PE and is another risk factor.
MICRO RNA DYSREGULATION POSTPARTUM
Women with a history of preeclampsia show similar dysregulation of micro RNA as women with CV disease (ref 98, 101, 109, 111). However, there are no data on micro RNA before pregnancies complicated by PE and thus it is not proven that PE is the cause of these alterations.
MICRO RNA DYSREGULATION IN OFFSPRING
Here again it remains unclear if alterations are due to underlying genetic abnormalities or epigenetic changes due to PE.
